# PROTOTYPE MATCHING NETWORKS FOR LARGE-SCALE MULTI-LABEL GENOMIC SEQUENCE CLASSIFICATION

## ABSTRACT

One of the fundamental tasks in understanding genomics is the problem of predicting Transcription Factor Binding Sites (TFBSs). With more than hundreds of Transcription Factors (TFs) as labels, genomic-sequence based TFBS prediction is a challenging multi-label classification task. There are two major biological mechanisms for TF binding: (1) sequence-specific binding patterns on genomes known as "motifs" and (2) interactions among TFs known as co-binding effects. In this paper, we propose a novel deep architecture, the Prototype Matching Network (PMN) to mimic the TF binding mechanisms. Our PMN model automatically extracts prototypes ("motif"-like features) for each TF through a novel prototype-matching loss. Borrowing ideas from few-shot matching models, we use the notion of support set of prototypes and an LSTM to learn how TFs interact and bind to genomic sequences. On a reference TFBS dataset with $2.1\ million$ genomic sequences, the PMN significantly outperforms baselines and validates our design choices empirically. To our knowledge, this is the first deep learning architecture that introduces prototype learning and considers TF-TF interactions for large scale TFBS prediction. Not only is the proposed architecture accurate, but it also models the underlying biology.

## 1 INTRODUCTION

Genomic sequences build the basis of a large body of research on understanding the biological processes in living organisms. Enabling machines to read and comprehend genomes is a long-standing and unfulfilled goal of computational biology. One of the fundamental task to understand genomes is the problem of predicting Transcription Factor Binding Sites (TFBSs), attracting much attention over the years (Consortium et al., 2012). Transcription Factors (TFs) are proteins which bind (i.e., attach) to DNA and control whether a gene is expressed or not. Patterns of how different genes expressed or not expressed control many important biological phenomena, including diseases such as cancer. Therefore accurate models for identifying and describing the binding sites of TFs are essential in understanding cells.

Owing to the development of chromatin immunoprecipitation and massively parallel DNA sequencing (ChIP-seq) technologies (Park, 2009), maps of genome-wide binding sites are currently available for multiple TFs in a few cell types across human and mouse genomes via the ENCODE (Consortium et al., 2012) database. However, ChIP-seq experiments are slow and expensive; they have not been performed for many important cell types or organisms. Therefore, computational methods to identify TFBS accurately remain essential for understanding the functioning and evolution of genomes.

An important feature of TFs is that they typically bind to sequence-specific patterns on genomes, known as "motifs" (Mitchell, 1989). Motifs are essentially a blueprint, or a "prototype" which a TF searches for in order to bind. However, motifs are only one part in determining whether or not a TF will bind to specific locations. If a TF binds in the absence of its motif, or it does not bind in the presence of its motif, then it is likely there are some external causes such as an interaction with another TF, known as co-binding effects in biology (Wang et al., 2012). This indicates that when designing a genomic-sequence based TFBS predictor, we should consider two modeling challenges: (1) how to automatically extract "motifs"-like features and (2) how to model the co-binding patterns and consider such patterns in predicting TFBSs. In this paper, we address both proposing a novel deep-learning model: prototype matching network (PMN).

To address **the first challenge** of motif learning and matching, many bioinformatics studies tried to predict TFBSs by constructing motifs using position weight matrices (PWMs) which best represented the positive binding sites. To test a sequence for binding, the sequence is compared against the PWMs to see if there is a close match (Stormo, 2000). PWM-matching was later outperformed by convolutional neural network (CNN) and CNN-variant models that can learn PWM-like filters Alipanahi et al. (2015a). Different from basic CNNs, our proposed PMN is inspired by the idea of "prototype-matching" (Wallis et al., 2008; Krotov & Hopfield, 2016). These studies refer to the CNN type of model as the "feature-matching" mode of pattern recognition. While pure feature matching has proven effective, studies have shown a "prototype effect" where objects are likely recognized as a whole using a similarity measure from a blurred prototype representation, and prototypes do not necessarily match the object precisely (Wallis et al., 2008). It is plausible that humans use a combination of feature matching and prototype matching where feature-matching is used to construct a prototype for testing unseen samples (Krotov & Hopfield, 2016). For TFBS prediction, the underlying biology evidently favors computation models that can learn "prototypes" (i.e. effective motifs). Although motifs are indirectly learned in convolutional layers, existing deep learning studies of TFBS (details in Section 3) have not considered the angle of "motif-matching" using a similarity measure. We, instead, propose a novel prototype-matching loss to learn prototype embedding automatically for each TF involved in the data.

None of the previous deep-learning studies for TFBS predictions have considered tackling **the second challenge** of including the co-binding effects among TFs in data modeling. From a machine learning angle, the genomic sequence based TFBS prediction is a multi-label sequence classification task. Rather than learning a prediction model for each TF (i.e., each label) predicting if the TF will bind or not on input, a joint model is ideal for outputting how a genomic sequence input is attached by a set of TFs (i.e., labels). The so-called "co-binding effects" connect deeply to how to model the dependency and combinations of TFs (labels). Multi-label classification is receiving increasing attention in deep learning (Wang et al., 2016; Guo & Gu, 2011; Wei et al., 2014) (detailed review in Section 3). Modeling the multi-label formulation for TFBS is an extremely challenging task because the number of labels (TFs) is in hundreds to thousands (e.g. 1,391 TFs in Vaquerizas et al. (2009)). The classic solution for multi-label classification using the powerset idea (i.e., the set of all subsets of the label set) is clearly not feasible (Tsoumakas & Katakis, 2006). Possible prior information about TF-TF interactions is unknown or limited in the biology literature.

To tackle these obstacles, our proposed model PMN borrows ideas from the memory network and attention literature. Vinyals et al. (2016) proposed a "matching network" model where they train a differentiable nearest neighbor model to find the closest matching image from a support set on a new unseen image. They use a CNN to extract features and then match those features against the support set images. We replace this support set of images with a learned support set of prototypes from the large-scale training set of TFBS prediction, and we use this support set to match against a new test sample. The key difference is that our PMN model is not for few-shot learning and we seek to *learn* the support set (prototypes). Vinyals et al. (2016) uses an attentionLSTM to model how a test sample matches to different items in the support set through softmax based attention. Differently, we use what we call a combinationLSTM to model how the embedding of a test sample matches to a combination of relevant prototypes. Using multiple "hops", the combinationLSTM updates the embedding of the input sequence by searching for which TFs (prototypes) are more relevant in the label combination. Instead of explicitly modeling interactions among labels, we try to use the combinationLSTM to mimic the underlying biology. The combinationLSTM tries to learn prototype embedding and represent high-order label combinations through a weighted sum of prototype embedding. This weighted summation can model many "co-binding effects" reported in the biology literature (Wang et al., 2012) (details in Section 2).

In summary, we propose a novel PMN model by combining few-shot matching and prototype feature learning. To our knowledge, this is the first deep learning architecture to model TF-TF interactions in an end-to-end model. In addition, this is also the first paper to introduce large scale prototype learning using a deep learning architecture. On a reference TFBS dataset with $2.1\ million$ genomic sequences, PMN significantly outperforms the state-of-the-art TFBS prediction baselines. We validate the learned prototypes through an existing database about TF-TF interactions. The TF groups obtained by clustering prototype embedding evidently captures the "cooperative effects" that has not been modeled by previous TFBS prediction works.

The main contributions of our model are:

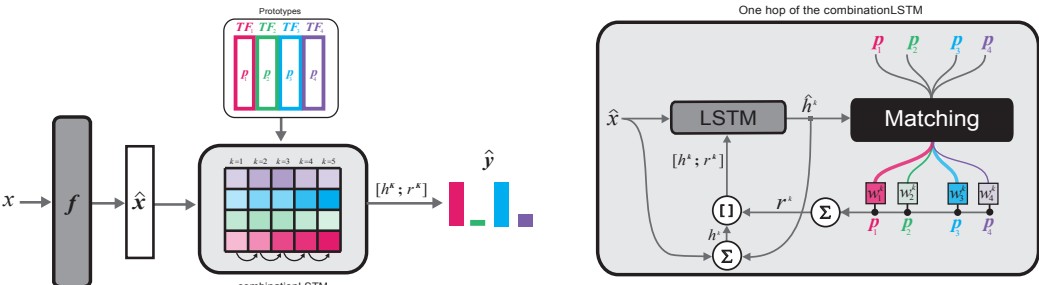

Figure 1: **Prototype Matching Network (PMN) Model**. On the left is an overview of the model. The input sequence $x$ is encoded as $\hat{x}$ using $f$ (3-layer CNN). $\hat{x}$ is then matched against the learned prototypes using the combinationLSTM for $K$ "hops" so that it can update its output based on TF interactions for this input sequence. The final output $\hat{y}$ is based on a concatenation of the final updated sequence vector $h^K$ from the LSTM, and final read vector $r^K$ from the matching. On the right is a closer look at the internal aspects of the combinationLSTM.

- We propose a novel model by combining few-shot matching with large-scale prototype feature learning.
- We design a novel prototype-matching loss to learn "motif"-like features in deep learning, which is important for the TFBS prediction task.
- We extend matching models from the few-shot single-label task to a large-scale multi-label task for genomic sequence classification.
- We implement an attention LSTM module to model label interactions in a novel way.
- Our model favors design choices mimicking the underlying biological processes. We think such modeling strategies are more fundamental especially on datasets from biology.

## 2 PROTOTYPE MATCHING NETWORKS

### 2.1 MODEL OVERVIEW

Given a DNA sequence $x$ (composed of characters A,C,G,T) of length $T$, we want to classify $x$ as a positive or negative binding site for each transcription factor $TF_1, TF_2, ..., TF_\ell$ in our dataset (i.e. multi-label binary classification). To do this, we seek to match $x$ to a bank of $\ell$ learned TF prototype vectors, $\{p_1, ..., p_\ell\}$, where each prototype is loosely representative of a motif. In addition, since TFs may bind or not bind based on other TFs, we model the interactions among TFs in order to make a prediction. An overview of our model can be seen in Figure 1.

### 2.2 EMBEDDING THE SEQUENCE AND PROTOTYPES

The input sequence $x \in \mathbb{R}^{4 \times t}$ is encoded using a function $f$ (3-layer CNN, which has shown to be sufficient for genomic feature extraction) to produce sequence embedding $\hat{x} \in \mathbb{R}^d$:

$$\hat{x} = f(x) \tag{1}$$

Each prototype vector $p_i \in \mathbb{R}^d$ is learned via a lookup table with a constant input integer at each position (e.g. 1 as input to the first position and $t$ as input to position $t$). I.e. the prototypes are produced by a multiplication of the identity matrix I and the learned lookup table matrix $W \in \mathbb{R}^{|TFs| \times d}$.

$$P = IW \tag{2}$$

Since our learned prototypes, $p_i$ are randomly initialized, we introduce a prototype matching loss $\mathcal{L}_p$, which forces a prototype to correspond to a specific TF. Our prototype matching loss is explained in section 2.4.

## 2.3 LSTM to learn Label Interactions and to Update the Sequence Embedding

Once we have the sequence and prototype embedding vectors, we want to compare the sequence to the prototypes to get binding site probabilities for each TF. The main idea is that we want to modify the sequence embedding $\hat{x}$ conditioned on matching against the prototypes. Since interactions among TFs influence binding, we cannot simply match the sequence to the prototypes. To obtain TF interactions, we use an LSTM (combinationLSTM), similar to the attention LSTM (attLSTM) in Vinyals et al. (2016). Our combinationLSTM is what does the actual matching for classification, whereas Vinyals et al. (2016) use the output of the attLSTM to make the final matching prediction. The combinationLSTM uses $K$ "hops" to process the prototypes $p_1, p_2, ..., p_\ell$ by matching against an updated sequence embedding $\hat{h}^k$. The hops allow the combinationLSTM to update the output vector based on which TFs match simultaneously. At each hop, the LSTM accepts a constant $\hat{x}$, a concatenation of the previous LSTM hidden state $\hat{h}^{k-1}$ and read vector r $r^{k-1}$, as well as the previous LSTM cell output $c^{k-1}$. $h^0$ and $c^0$ are initialized with zeros, and $r^0$ is initialized with the mean of all prototype vectors, $\frac{1}{|p|} \sum_i^{|p|} p_i$.

The output hidden state $\hat{h}^k$ is matched against each prototype using cosine similarity, producing a similarity score. Since this similarity is in the range [-1,1], we feed this output through sigmoid function, weighted by hyperparameter $\epsilon$ (we use $\epsilon$=20) to produce the similarity score $w_i^k$ at hop $k$ in [0,1]. The read vector $r$ is updated by a weighted sum of the prototype vectors using the matching scores. At each hop, $h^k$ is updated using the current LSTM output hidden state and the sequence embedding.

$$\hat{h}^k, c^k = \text{LSTM}(\hat{x}, [h^{k-1}; r^{k-1}], c^{k-1}) \tag{3}$$

$$h^k = \hat{h}^k + \hat{x} \tag{4}$$

$$r^k = \sum_{i=1}^{|p|} w_i^k p_i \tag{5}$$

$$w_i^k = 1/\left(1 + e^{-\epsilon c(\hat{h}^k, p_i)}\right) \tag{6}$$

$$c(u, v) = \frac{u \cdot v}{||u||_2 ||v||_2} \tag{7}$$

In the TFBS task, eq. 5 is the important factor for modelling TF combinations. The output vector $r$ can model multiple prototypes matching at once through a linear combination. Furthermore, the LSTM with $K$ hops is needed because of the fact that a TF binding may influence other TFs in a sequential manner. For example, if $TF_i$ matches to $\hat{h}^k$ in the first hop, $r^k$ is then used to output $\hat{h}^{k+1}$ which can match to $TF_j$ at the next hop. In this case, $\hat{h}^{k+1}$ is a joint representation of $\hat{x}$ and the current matched prototypes, represented by $r^k$. At each hop, the LSTM fine-tunes $w^k$ in order to find $TF$ binding combinations.

The final output $\hat{y} \in \mathbb{R}^{|TFs|}$ is computed from a concatenation of the final hidden state and read vectors $[h^k; r^K]$ after the $K^{th}$ hop using a linear transform and an element-wise sigmoid function to get a probability of binding for each TF. :

$$o = W([h^K; r^K]) \tag{8}$$

$$\hat{y} = 1/(1 + e^{-o}) \tag{9}$$

## 2.4 Classification and Prototype Matching Loss Functions

To classify a sequence, we use a standard binary cross entropy loss between each label $y_i$ for $TF_i$, and the corresponding $TF_i$ output $\hat{y}_i$, which we call the classification loss, $\mathcal{L}_c$, for each label.

We also introduce a prototype matching loss $\mathcal{L}_p$, which forces a prototype to correspond to a specific TF since prototypes are learned from random initializations. The prototype matching loss works by using an L$_2$ between the true label $y_i$ for $TF_i$ and the final matching weight $w_i^K$ between updated sequence $h^K$ and prototype $p_i$. This loss forces a prototype to match to all of its positive binding sequences. Each $w_i^K$ is from the final prototype matching weights after the $K^{th}$ hop from Eq (6).

Table 1: Comparison of previous deep-learning studies for TFBS and three closely related deep learning papers in the recent literature. The columns indicate properties: (1) whether the study has a joint deep architecture for multi-label prediction or not, (2) if the study learns prototype features ("motifs" in the TFBS literature), (3) whether the study models how input samples match prototypes, (4) if it uses RNN to model high-order combinations of labels, and finally (5) if the method considers current sample inputs for modeling label combinations. All previous TFBS studies do not model label interactions. PMN combines several key strategies from deep learning literature including: (a) learning label-specific prototype embedding (Snell et al., 2017) through prototype-matching loss, (b) using RNN to model higher-order label combinations (Wang et al., 2016), and (c) using LSTM to model such combinations dynamically (conditioned on the current input) (Vinyals et al., 2016). PMN is the only model that exhibits all desirable properties.

| Method | Multi-Label Joint Model | Task | Prototypes (Motifs) per Label | Prototype Matching Loss | RNN for Label Combinations (Co-Binding) | Dynamic Label Combinations |
|---|---|---|---|---|---|---|
| DeepBind (Alipanahi et al., 2015a) | × | TFBS | × | × | × | × |
| DeepSEA (Zhou & Troyanskaya, 2015) | ✓ | TFBS | × | × | × | × |
| DanQ (Quang & Xie, 2016) | ✓ | TFBS | × | × | × | × |
| TFImpute(Qin & Feng, 2017) | ✗[1] | TFBS | ✓ | × | × | × |
| CNN-RNN (Wang et al., 2016) | ✓ | Image | ✓ | × | ✓ | × |
| Memory-Matching (Vinyals et al., 2016) | × | FewShot | × | ✓ | ✗[2] | ✗[2] |
| Prototypical-Network (Snell et al., 2017) | × | FewShot | ✓ | × | × | × |
| Prototype Matching Network: (this paper) | ✓ | TFBS | ✓ | ✓ | ✓ | ✓ |

The important thing is that the loss is computed from the final weights $w^K$. This allows the LSTM to attend to certain TFs at different hops before making its final decision, modeling the co-binding of TFs. The hyperparameter $\lambda$ controls the amount that each prototype is mapped to a specific TF. $\lambda=0$ corresponds to random prototypes since we are not forcing $p_i$ to match to a specific sequence.

Thus, the final loss $\mathcal{L}$ is a summation of both the classification loss and the prototype matching loss:

$$\mathcal{L} = -\mathcal{L}_c - \lambda\mathcal{L}_p \tag{10}$$

$$\mathcal{L}_c = \sum_i^{|p|}(y_i\log\hat{y}_i + (1 - y_i)\log(1 - \hat{y}_i)) \tag{11}$$

$$\mathcal{L}_p = \sum_i^{|p|}(y_i - w_i^K)^2 \tag{12}$$

## 2.5 TRAINING

We trained our model using Adam (Kingma & Ba, 2014) with a batch size of 512 sequences for 40 epochs. Our results were based on the test set results from the best performing validation epoch. We use dropout (Srivastava et al., 2014) for regularization.

## 3 RELATED WORKS

**Deep learning in bioinformatics:** Deep learning is steadily gaining popularity in the bioinformatics community. This trend is credited to their ability to extract meaningful representations from large datasets. For instance, multiple recent studies have successfully used deep learning for modeling protein sequences (Lin et al., 2016; Zhou & Troyanskaya, 2014), modeling DNA sequences (Alipanahi et al., 2015b; Lanchantin et al., 2016), predicting gene expression (Singh et al., 2016), as well as understanding the effects of non-coding variants (Zhou & Troyanskaya, 2015; Quang & Xie, 2016)).

**Previous Studies of TFBS and more:** Previous techniques for predicting TFBS include many sequence-motif based computational approaches that typically use position-based sequence information (Stormo, 2000). Relying on a set of known transcription factor binding sites (TFBSs) for a given

---

[1]The original TFImpute model is not multi-label. A modified version for multi-label learning with auxilary data was used to compare to Zhou & Troyanskaya (2015)

[2]The memory matching network (Vinyals et al., 2016) uses the attention LSTM to model the dependency among items in a support set dynamically (conditioned on current input sample).

TF, the binding preference is generally represented in the form of a position weight matrix (PWM) (Stormo, 2013; Mathelier et al., 2013) (also called position-specific scoring matrix) derived from a position frequency matrix (PFM). Recently this technique was outperformed by different variations of deep convolutional models (Alipanahi et al., 2015b; Lanchantin et al., 2016; Quang & Xie, 2016; Shrikumar et al., 2017). While motif-based PWMs are compact and interpretable, they can under-fit ChIP-seq data by failing to capture subtle but detectable and important sequence signals, such as direct DNA-binding preferences of certain TFs, cofactor binding sequences, accessibility signals, or other discriminative sequence features (Arvey et al., 2012; Le et al., 2017).

Table 1 summarizes four most relevant deep learning studies of TFBS in the literature. Alipanahi et al. (2015b) was the first to use a deep learning based approach to predict TFBSs. They showed that a 1-layer CNN could outperform baseline motif matching approaches which used position weight matrices Machanick & Bailey (2011). Zhou & Troyanskaya (2015) used a similar method to predict the effects of variants in noncoding regions of the genome, where TFBS prediction was an intermediate step to predict variant effects. They used a 3-layer CNN model to predict 919 chromatin labels (including 690 TFBS labels). Quang & Xie (2015) extended this model using a bidirectional LSTM on top of the CNN outputs to model interactions among motifs. Note that this is not modeling interactions among labels (TFs), but rather among sequence features. Qin & Feng (2017) uses a similar lookup table approach for learning a representation for a TF, but their implementation is for transferring between cell lines where the target cell line has no experimental data. However, ChIP-seq experiments are relatively cheap given a specific TF and sequence of interest. We are more interested in modeling the underlying biology.

Aside from deep learning models, there have been other works to model joint dependencies among TFs Kazemian et al. (2011); He et al. (2009); Gautier et al. (2008); Sinha & He (2007). Our primary goal is to extend previous deep learning methods to better model co-binding, thus we do not compare against these approaches.

The formulation of TFBS prediction belongs to a general category of "biological sequence classification". Sequence analysis plays an important role in the field of bioinformatics. Various methods have previously been proposed, including generative (e.g., Hidden Markov Models-HMMs) and discriminative approaches. Among the discriminative approaches, string kernel methods provide some of the most accurate results, such as for remote protein fold and homology detections (Leslie & Kuang, 2004; Kuksa et al., 2008). We omit a full survey of this topic due to its vast body of previous literature and loose connection to our TFBS formulations.

**Memory Matching Network and Attention RNN:** Attention in combination with RNNs has been successfully used for many tasks Bahdanau et al. (2014). Various methods have extended the single step attention approach to attention with multiple 'hops' Sukhbaatar et al. (2015); Kumar et al. (2016). For example, for the Question Answering task, Kumar et al. (2016) introduced the Dynamic Memory Network which uses an iterative attention process coupled with an RNN over multiple 'episodes'. Kumar et al. (2016) show that the attention gets more focused over successive hops or 'episodes' over the input, reaffirming the need for recurrent episodes. Most of these models are for sequential inputs. For tasks that involve a set (i.e., no order) as input or/and as output, Vinyals et al. (2015) introduced a general framework employing a similar content based attention with associative memory. To deal with non-sequential inputs, the input elements are stored as a unordered external memory. It uses an LSTM coupled with a dynamic attention mechanism over the memory vectors. After 'K' hops of the LSTM, the final output/memory retrieved from the process block does not depend on the ordering of the input elements. Vinyals et al. (2016) leverage this set framework for few-shot learning by introduce the "matching network" (MN) model. The MN model learns nearest neighbor classifier to find the closest matching image from a support set on a new unseen image. To this end, they use the same 'process' block from Vinyals et al. (2015) with modifications to incorporate 'matching'. In each hop over the support set, they 'match' or compare the hidden state of the attLSTM with each of the support set elements. They use the output of the attLSTM to do the final support set matching.

In our model, we use a set of learned prototype vectors instead of the support set of images. Both our model and the MN model uses an LSTM to learn the interactions among the items in the support set. However, the MN model uses a softmax attention and we use a sigmoid attention (due to the multi-label output). We compare a baseline model using a softmax attention (details in section 4.1).

Table 2: Comparison to similar matching models

|                      | Task        | Output      | Comparison         | Support Set                         |
|----------------------|-------------|-------------|--------------------|-------------------------------------|
| **Vinyals et al. 2015** | Few shot    | Single task | Cosine Similarity  | Individual support set images       |
| **Snell et al. 2017**   | Few shot    | Single task | Squared Euclidean  | Mean of each class from support set |
| **Ours**                | Large-scale | Multi-task  | Cosine Similarity  | Learned prototype for each class    |

**Prototype Features and Prototypical Networks:** While standard deep learning architectures have proven to work in many tasks, most operate under the feature-matching theory of pattern recognition where an input is decomposed into a set of features, and then are compared with those stored in the memory (Krotov & Hopfield, 2016). Prototype theory, on the other hand, proposes that objects are recognized as a whole and prototypes do not necessarily match the object precisely (Wallis et al., 2008). In this sense, the prototypes are blurred abstract representations which include all of the object's features. Krotov & Hopfield (2016) show that pattern recognition is likely a combination of both feature-matching and prototype-matching. Transcription factors bind to motifs on DNA sequences, which we view as prototypes, the blurred features are constructed from a CNN (feature-matching). Our method is motivated by the prototype-matching theory, where instead of searching for exact features to match against, the model tests an unseen sample against a set of prototypes using a defined similarity metric to make a classification.

Snell et al. (2017) introduces prototypical networks for zero and one-shot learning, which assumes that the data points belonging to a particular class cluster around a single prototype. This prototype is representative for its class. In this model, each prototype embedding is the average embedding of all examples belonging to a certain class. This method is equivalent to a linear classifier if the Euclidean distance metric is used. While this method successfully learns a single prototype embedding for each class, it does not utilize a recurrent-attention mechanism over the support set. The prototypes in their method are restricted to the average embedding of its class and do not consider interactions among classes. Instead, our prototypes are the learned embedding for each label and play important roles in modelling the interactions among labels. Rippel et al. (2015) proposed a method for learning k-prototypes within each class, instead of just one. This in turn requires a k-means classifier to first group the intra-class embeddings before separating inter-class embeddings.

**Multi-label Classification in Deep Learning:** Multi-label classification is receiving increasing attention in image classification and object recognition. In a multi-label classification task, multiple labels may be assigned to each instance. A problem transformation method for multilabel classification considers each different set of labels as a single label. Thus, it learns a binary classifier for every element in the powerset of the labels (Tsoumakas & Katakis, 2006). However, this may not be feasible in the case of a large number of labels. The most common and a more feasible problem transformation method learns a binary classifier for each label(Tsoumakas & Katakis, 2006).Gong et al. (2013) use ranking to train deep convolutional neural networks for multi-label image classification. In Hypotheses-CNN-Pooling (Wei et al., 2014), the output results from different hypotheses are aggregated with max pooling. These models treat the labels independently from each other. However, modeling the dependencies or co-occurrences of the labels is essential to gain a complete understanding of the image. To this end, Read et al. (2009) propose a chaining method to model label correlations. Xue et al. (2011), Ghamrawi & McCallum (2005) and Guo & Gu (2011) use graphical models to capture these dependencies. However, these approaches only model low order label correlations and can be computationally expensive. Wang et al. (2016) is the state-of-the-art multi-label study for object recognition. To characterize the high-order label dependencies, this model leverages the ability of an LSTM to model long-term dependency in a sequence. Briefly, the label prediction is treated as an 'ordered prediction path'. This prediction path is essentially modeled by the RNN. While the CNN is used to extract image features, the RNN model takes the current label prediction as input at each time step and generates an embedding for the next predicted label. Although the RNN utilizes its hidden state to model the label dependencies, it is not dynamically conditioned on input samples. StarSpace Wu et al. (2017) is a method to learn entity embeddings in the input space which can handle multi-label outputs. Our method is different in that it extracts relationships directly among the outputs rather than in the input space.

Table 3: Dataset Overall Summary. In each split, about 40% of its samples have more than 1 TF binding (i.e. a combination of TFs binding together), and each sample has an average of about 5.7 TFs binding.

| Split | Total Samples | Co-Binding Samples | Mean # of TFs Binding Per Sample |
|---|---|---|---|
| Train | 1446320 | 797475 | 5.62 |
| Valid | 331884 | 186509 | 5.75 |
| Test | 306297 | 170329 | 5.85 |

Figure 2: Dataset Per-TF Summary. For each TF, the left y-axis shows the percentage of positive samples this TF has out of all possible samples. For each TF, the right y-axis shows the percentage of positive samples which also have another TF binding.

## 4 EXPERIMENTS

### 4.1 DATASETS AND EXPERIMENTAL SETUP FOR TFBS PREDICTION

**Dataset:** We constructed our own dataset from ChIP-seq experiments in the ENCODE project database (Consortium et al., 2012). ChIP-seq experiments give binding affinity p-values for certain locations in the human genome for a specific cell type. We divided the entire human genome up into 200-length sequences, using a sliding window of 50 basepairs. We then extracted the 200-length windows surrounding the peak locations for 86 transcription factors in the human lymphoblastoid cell line (GM12878). Any peak with a measured p-value of at least 1 was considered a positive binding peak (it is important to note that we could get better results by setting a higher treshold, but we were interested in modelling all potential peaks). For each window in the genome, if any of the TF windows have a >50% overlap, we consider this a positive binding site window. We discard all windows with no TFs binding, resulting in a total of 2,084,501 binding site windows, or about 14% of the human genome.

**Data Statistics:** We use windows in chromosomes 1, 8, and 21 as a validation set (331,884 sequences), chromosomes 3, 12, and 17 as a test set (306,297 sequences), and the rest as training (1,446,320 sequences). The number of positive training windows for each TF ranges from 793 (<1% of training samples) to 380,824 (23% of training samples). The validation and test splits have similar percentages. About 40% of the windows have more than 1 TF binding to it, and each window has about 5 TFs binding. A overview summary of the dataset is shown in Table 3 and a per-TF summary of the dataset is shown in Figure 2.

**Model Variations:** To test the PMN model on our TFBS dataset, we constructed 3 model variations

1. **CNN** As commonly used in previous models (Quang & Xie, 2015; Lanchantin et al., 2016), we use a baseline 3-layer CNN model. We use {512,256,128} kernels of widths {9,5,3} at the 3 layers, respectively. The output of the CNN is maxpooled across the length, resulting in a final output vector of size 128. This architecture is used for both the single-label and multi-label CNN models.

Table 4: TFBS Prediction Across 86 TFs in GM12878 Cell Line. We compare our PMN model to two CNN baseline models (single-label and multi-label). We use the same CNN model and extend it using prototypes and the combinationLSTM to create our PMN model. $\lambda$ represents the weighting of the prototype loss in eq. 10. Results are shown using statistics across all 86 TFs, where our PMN model outperforms the CNN models based on all 3 metrics used. The PMN also outperforms both CNN models significantly using a pairwise t-test.

| Model | auROC | | | auPR | | | Recall at 50% FDR | | |
|---|---|---|---|---|---|---|---|---|---|
| | Mean | Std. | % Increase over single | Mean | Std. | %Increase over single | Mean | Std. | %Increase over single |
| CNN (single-label) | 0.820 | 0.072 | - | 0.263 | 0.123 | - | 0.224 | 0.198 | - |
| CNN (multi-label) | 0.831 | 0.055 | 1.37 | 0.257 | 0.113 | -2.52 | 0.215 | 0.186 | -4.00 |
| PMN ($\lambda$=1), no LSTM | 0.830 | 0.057 | 1.30 | 0.267 | 0.116 | 1.22 | 0.231 | 0.197 | 3.09 |
| PMN ($\lambda$=1), softmax att | 0.834 | 0.057 | 1.70 | **0.272** | 0.115 | **3.36** | **0.243** | 0.194 | **8.48** |
| PMN ($\lambda$=0), sigmoid att | 0.837 | 0.055 | 2.13 | 0.271 | 0.113 | 3.00 | 0.229 | 0.186 | 1.92 |
| PMN ($\lambda$=0.5), sigmoid att | 0.839 | 0.055 | 2.38 | **0.272** | 0.113 | **3.36** | 0.235 | 0.187 | 4.73 |
| PMN ($\lambda$=1), sigmoid att | **0.840** | 0.054 | **2.45** | 0.270 | 0.114 | 2.47 | 0.234 | 0.187 | 4.17 |

2. **PMN no LSTM** To demonstrate the effectiveness of the combinationLSTM module, we implement a PMN with no LSTM to iteratively update its weightings. In this model, we use eq. 5-9, except that we replace $\hat{h}^k$ in eq. 9 with $\hat{x}$ since there is no LSTM. The output (eq. 11) is then a concatenation of $r$ and $\hat{x}$. We still use the full prototype loss ($\lambda = 1$) in this model.

3. **PMN, softmax att** Since softmax attention is typically used in attention models, we explored using a softmax function to replace eq. 6 from $k = 0$ until $k = K - 1$, and then an element-wise sigmoid function (i.e. eq. 6) for the final output since it is multi-label classification.

4. **PMN, sigmoid att** The full PMN model utilizes the LSTM module in eq. 6 over $K$ hops. We implement 3 variations of the prototype loss ($\lambda = 0, \lambda = 0.5, \lambda = 1$), where $\lambda = 0$ represents no prototype loss, or random prototypes. We observed that $\lambda > 1$ did not result in improved results.

We then extended the baseline CNN to use the learned prototypes $p_i$, and prototype matching LSTM (combinationLSTM). We call the combination of the CNN, prototypes, and combinationLSTM a PMN. We use $K = 5$ hops for the combinationLSTM because each sample has on average 5 positive label outputs. We also compared against a baseline single-task model for each TF, which assumes no interactions among TFs.

**Metrics:** We use three separate metrics which are commonly used in large scale TFBS prediction. Since our labels are very unbalanced, we use area under ROC curve (auROC). However, auROC may not give a fair evaluation in unbalanced datasets (Lever et al., 2016; Ching et al., 2017). So we also use area under precision-call curve (auPR), and recall at 50% false discovery rate.

## 4.2 Large-Scale TFBS Classification Results

Table 4 shows the results of our models across the 86 TF labels. The joint CNN (multi-label) model outperformed the single label CNN models in auROC and auPR. The main advantage of the joint model is that it is faster than an individual model for each TF. The joint model's improvement over the single-task models was not significant (p-value $< 0.05$) based on a one-tailed pairwise t-test. This is presumably because the joint model finds motifs similar among all motifs, but it doesn't model interactions among TF labels.

The PMN model outperformed both baseline CNN models in all 3 metrics. In addition, the improvement of the PMN over both CNN models was significant using a one-tailed pairwise t-test. We hypothesize that the combinationLSTM module accurately models co-binding better, leading to an increase in performance.

In Figure 3, we show the per-epoch mean auROC results of the PMN vs CNN model. There are two important factors to note from this plot. First, the PMN models all outperform the baseline CNN. Second, the PMN models converge faster than the CNN. We hypothesize that the prototypes and similarity measure help the model generalize quickly. We assume this is the case since prototype matching models have been shown to work in few-shot cases (Vinyals et al., 2016; Snell et al., 2017).

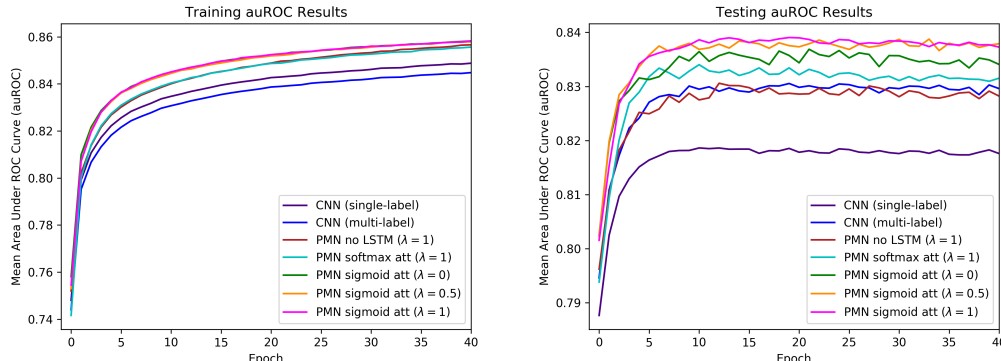

Figure 3: **TFBS Per-Epoch Mean auROC** (Left: Train, Right: Test)

Table 5: Column "TF-A" represents a TF in one of the clusters obtained from hierarchical clustering. The subsequent column contains TFs that belong to the *same cluster* and, according to TRRUST database (Han et al., 2015), share target genes with TF-A. Remaining columns show the number of overlapping target genes between each TF and TF-A pair as well as their p-values for TF-TF cooperativity obtained from TRRUST. An interesting thing to note here is that in Cluster 1 and 2, TF that is away from TF-A in the cluster has lower p-value than the TF that is closer.

| Cluster | TF-A | TFs sharing targets | No. of target genes | P-value |
|---|---|---|---|---|
| 1 | | EP300 | 9 | 1.08E-10 |
| | MYC | USF1 | 4 | 5.77E-04 |
| | | E2F4 | 3 | 3.93E-04 |
| 2 | | PAX5 | 8 | 8.86E-10 |
| | RELA | ATF2 | 2 | 6.10E-04 |
| | | CREBP | 2 | 5.04E-04 |
| 3 | ATF2 | CREB1 | 8 | 2.69E-12 |
| 4 | YY1 | MAF | 2 | 7.03E-04 |
| 5 | STAT5A | BRCA1 | 2 | 8.54E-04 |
| 6 | MAX | STAT3 | 2 | 9.47E-04 |

This is also validated in TFs with the smallest amount of samples. In the 10 TFs with the smallest amount of samples, the PMN ($\lambda = 1$) has a 1.86% increase in mean auROC over the CNN. In the 10 TFs with the largest amount of samples, however, the PMN only results in an increase of 0.94%.

**Biological Validation of Learned Prototype Embedding:** In our PMN model, we are learning a prototype for each TF that represents "motif"-like features for that TF. Each prototype is processed by the combinationLSTM which is capturing the information about TFs that bind simultaneously (or TF-TF cooperativity). Thus, we hypothesize that our prototype not only learns its TF embedding but should also reflect the TF-TF cooperativity. This information is biologically relevant as it answers a critical question in the field - "What TFs work together (or cooperate) to regulate a particular gene of interest?". To test this hypothesis, we performed hierarchical cluster analysis on the prototypes for the 86 TFs and obtained multiple clusters containing 2 or more TFs (Figure in Appendix). Next, we searched TFs from each cluster in a reference database of human transcriptional regulatory interactions called TRRUST (Han et al., 2015). For each TF, say "TF-A", in its database, TRRUST displays a list of TFs that regulate the same target genes as TF-A. It also shows the measures of the significance of their cooperativity as p-values, using protein-protein interactions derived from major databases. Having no expert knowledge in biology, we found some interesting results that are summarized in Table 5. We found pairs of TFs (in Clusters 1-6), whose prototypes had been clustered together, to have significant (p-value < 0.0001) cooperativity curated in TRRUST database. These observations indicate that each prototype is learning sufficient combinatorial information that allows the clustering algorithm to group the TFs that cooperate during gene regulation. Therefore, the learned prototypes are not only guiding the model to better predictions but are capturing an embedding that can provide insights into the TF-TF cooperativity in the actual biological scenario.

## 5   CONCLUSION

Sequence analysis plays an important role in the field of bioinformatics. A prominent task is to understand how Transcription Factor proteins (TFs) bind to DNA. Researchers in biology hypothesize

that each TF searches for certain sequence patterns on genome to bind to, known as "motifs". Accordingly we propose a novel prototype matching network (PMN) for learning motif-like prototype features. On a support set of learned prototypes, we use a combinationLSTM for modeling label dependencies. The combinationLSTM tries to learn and mimic the underlying biological effects among labels (e.g. co-binding). Our results on a dataset of $2.1\ million$ genomic strings show that the prototype matching model outperforms baseline variations not having prototype-matching or not using the combinationLSTM. This empirically validates our design choices to favor those mimicking the underlying biological mechanisms.

Our PMN model is a general classification approach and not tied to the TFBS applications. We show this generality by applying it on the MNIST dataset and obtain convincing results in Appendix Section 7.1. MNIST differs from TFBS prediction in its smaller training size as well as in its multi-class properties. We plan a few future directions to extend the PMN. First, TFBSs vary across different cell types, cell stages and genomes. Extending PMN for considering the knowledge transfer is especially important for unannotated cellular contexts (e.g., cell types of rare diseases or rare organisms). Another direction is to add more domain-specific features. While we show that using prototype matching and the combinationLSTM can help modelling TF combinations, there are additional raw feature extraction methods that we could add in order to obtain better representations of genomics sequences. These include reverse complement sequence inputs or convolutional parameter sharing (Shrikumar et al., 2017), or an RNN to model lower level spatial interactions (Quang & Xie, 2015; Lanchantin et al.) among motifs.

## 6 ACKNOWLEDGEMENTS

We would like to thank Dr. Chongzhi Zang from the University of Virginia Medical School for helping generate the datasets and for the helpful discussions.

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

# 7 APPENDIX

## 7.1 MNIST

To validate the learned prototype matching on another task, we compare our PMN model against a standard CNN model on the MNIST dataset. We use a 3-layer CNN with {32,32,32} kernels of sizes {5,5,5} at the 3 layers, respectively.

For the MNIST experiments, the PMN models do not show a drastic improvement over the baseline CNN. The results are shown in Table 6. However, we do note that based on the per-epoch plots in Figure 4, the PMN models do converge faster than the baseline CNN. In addition, we show in figure 5 that the PMN embeddings are better separated than the CNN embeddings. This is likely due to the fact that the PMN uses a similarity metric, and the fact that it can update its embedding based on which number prototypes it matches to. In other words, if an image looks similar to several numbers (e.g. "5" and "6"), the PMN can update its output based on which one it matches more to. Note that although we train using a prototype loss for each class, we do not constrain the matching to only match to one prototype.

Table 6: MNIST using 3-Layer CNN

| Model | Accuracy |
|---|---|
| **CNN** | 99.37 |
| **PMN** ($\lambda = 0$) | 99.43 |
| **PMN** ($\lambda = 0.5$) | 99.39 |
| **PMN** ($\lambda = 1$) | **99.50** |

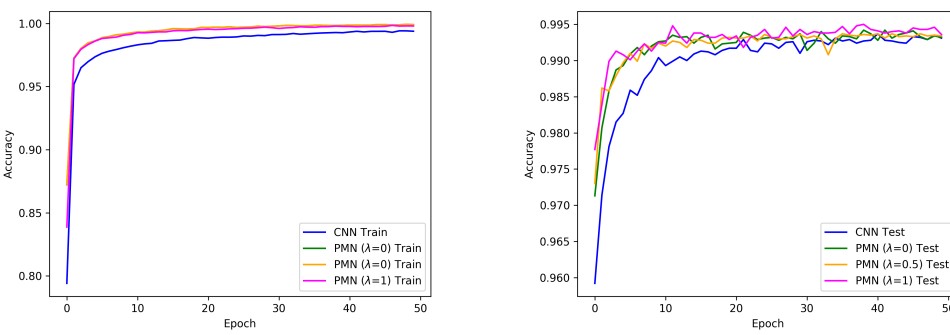

Figure 4: **MNIST Per-Epoch Accuracy** (Left: Train, Right: Test)

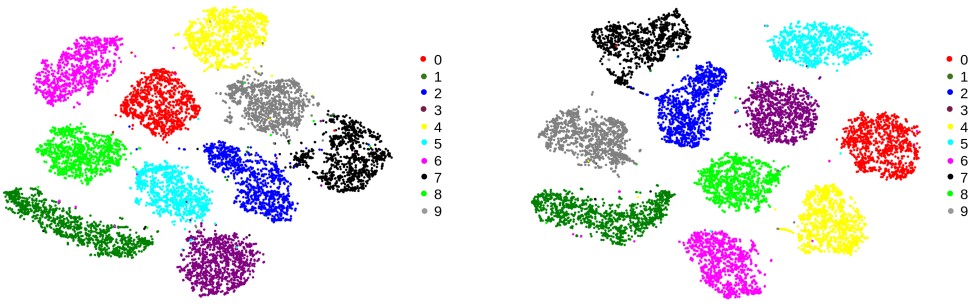

Figure 5: 2d T-SNE Embedding of final output vector before classification. Left: CNN, Right: PMN

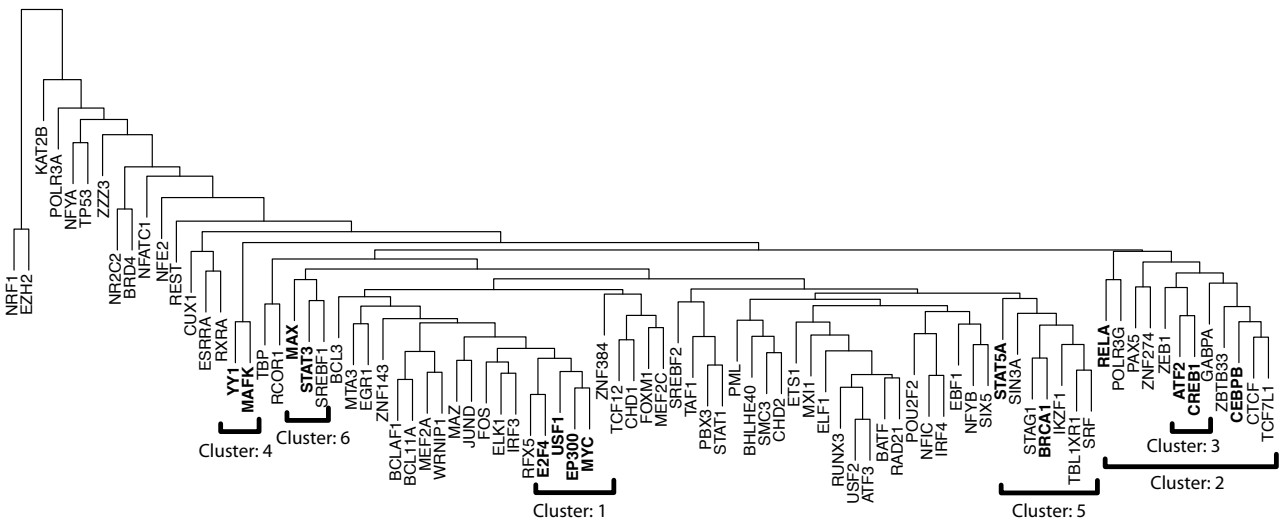

Figure 6: **Hierarchical clustering of prototypes of 86 TFs.**

