# OpenReview forum: "Prototype Matching Networks for Large-Scale Multi-label  Genomic Sequence Classification"
_ICLR.cc/2018/Conference — Reject_

### Official Review · AnonReviewer2 · 2017-11-24
**This work may have some value in applications, but contains lots of issues.**

**Rating:** 5
**Confidence:** 5

**Review:**

The authors of this manuscript proposed a model called PMN based on previous works for the classification of transcription factor binding. Overall, this manuscript is not well written. Clarification is needed in the method and data sections. The model itself is an incremental work, but the application is novel. My specific concerns are given below.

1. It is unclear how the prototype of a TF is learned. Detailed explanation is necessary.

2. Why did the authors only allow a TF to have only one prototype? A TF can have multiple distinct motifs.

3. Why peaks with p-value>=1 were defined as positive? Were negative classes considered in the computational experiments?

4. What's the relationship between the LSTM component in the proposed method and sparse coding?

5. The manuscript contains lots of low-end issues, such as:
5.1. Inconsistency in the format when referring to equations (eq. equation, Equation, attention LSTM, attentionLSTM, t and T etc);
5.2. Some "0"s are missing in Table 3;
5.3. L2 should be L_2 norm;
5.4. euclidean -> Euclidean; pvalue-> p-value;
5.5. Some author name and year citations in the manuscript should be put in brackets;
5.6. The ENCODE paper should be cited properly, ("Consortium et al., 2012" is weird!) ;
5.7. The references should be carefully reformatted, for example, some words in the references should be in uppercase (e.g. DNA, JASPER, CNN etc.), some items are duplicated, ...

Comments for the revised manuscript: I decide to keep my decision as it is. My major and minor concerns are not fully well addressed in the revised paper.

---

> ### Author Response · Authors · 2018-01-05
> **Response to 5.**
>
> We thank the reviewer for pointing out the low-end issues. We have since fixed these in the manuscript.

---

> ### Author Response · Authors · 2018-01-05
> **Response to 4.**
>
> The LSTM component is not connected to the sparse coding method. The proposed LSTM aims to model the dependency among labels.

---

> ### Author Response · Authors · 2018-01-05
> **Response to 3.**
>
> We realize that we did not explain the dataset construction very well. We have updated our explanation.

---

> ### Author Response · Authors · 2018-01-05
> **Response to 2.**
>
> It is true that a TF may have multiple motifs. However, it is assumed that the secondary motifs are in fact primary motifs of other TFs (Wang et al. 2012). In addition, as pointed out by Snell et al. 2017, Multiple prototypes were proposed in Mensink et al. and Rippel et al. But, both methods require a separate partitioning phase that is decoupled from the weight updates, which complicates the model. We found that adding additional prototypes made training more difficult, and did not improve the accuracy.

---

> ### Author Response · Authors · 2018-01-05
> **Response to 1.**
>
> We use a lookup table for representing TFs’ prototypes. This means for each TF  we learn an embedding vector representing this TF’s pattern. The lookup table is learned and updated via gradient descent on each update to minimize the prototype-matching loss we proposed in the paper. We thank the reviewer for noting this problem, and we have updated the manuscript to better explain the lookup table.

---

### Official Review · AnonReviewer3 · 2017-11-28
**This paper presents a method to learn motifs and cooperative binding structure from data for multiple TF binding (ChIP-Seq).**

**Rating:** 5
**Confidence:** 3

**Review:**

Summary
This paper proposes a prototype matching network (PMN) to model transcription factor (TF) binding motifs and TF-TF interactions for large scale transcription factor binding site prediction task. They utilize the idea of having a support set of prototypes (motif-like features) and an LSTM from the few shot learning framework to develop this prototype matching network. The input is genomic sequences from 14% of the human genome, each sequence in the dataset is bound by at least one TF. First a Convolutional Neural Network with three convolutional layers is trained to predict single/multiple TF binding. The output of the last hidden layer before sigmoid transformation is used as the LSTM input. A weighted sum of similarity score (sigmoid of cosine similarity, similar to attention mechanism of LSTMs) along with prototype vectors are used to update the read vector. The final output is a sigmoid of the final hidden state concatenated with the read vector. The loss function used is difference of a cross-entropy loss function and a lambda weighted prototype loss function. The latter is the mean square error between the output label and the similarity score.  The authors compare the PMN with different lambda values with CNN with single/multi-label and see marginal improvement in auROC, auPR and Recall at 50% FDR with the PWM. To test that PWN finds biologically relevant TF interactions, the authors perform hierarchical clustering on the prototypes of 86 TFs and compare the clusters found to the known co-regulators from the TRRUST database and find 6 significant clusters.


Pros:
1. The authors utilize the idea of prototypes and few shot learning to the task of TF-binding and cooperation.

2. Attention LSTMs are used to model label interactions.

Just like CNN can be related to discriminative training of PSSM or PWM, the above points demonstrate nicely how ideas/concepts from the recent developments in DL can be adopted/relate (and possibly improve on) to  similar generative modeling approaches used in the past  for learning cooperative TF binding.

Cons:

1. Authors do not compare their model’s performance to the previously published TF binding prediction algorithms (DeepBind, DeepSEA).
2. The authors miss important context and make some inaccurate statements: TF do not just “control if a gene is expressed or not” (p.1). It’s not true that previous DL works did not consider co-binding. Works such as DeepSea combined many filters which can capture cooperative binding to define which sequence is “regulated”. It is true this or DeepBind did not construct a structure a structure over those as learned by an LSTM. The authors do point out a model that does add LSTM (Quang and Xie) but then do not compare to it and make a vague claim about it modeling interactions between features but not labels (p. 6 top). Comparing to it and directly to DeepSee/Bind seems crucial to claim improvements on previous works. Furthermore, the authors acknowledge the existence of vast literature on this specific problem but completely discard it as “loose connection to our TFBS formulation”. In reality though, many works in that area are highly relevant and should be discussed in the context of what the authors are trying to achieve. For example, numerous works by Prof. Saurabh Sinha have focused specifically on joint TF modeling (e.g. Kazemian NAR 2011, He Plos One 2009, Ivan Gen Bio 2008, MORPH Plos Comp Bio 2007). In general, trying to lay claims about significant contributions to a problem, as stated here by the authors, while completely disregarding previous work simply because it’s not in a DL framework (which the authors are clearly more familiar with) can easily alienate reviewers and readers alike.

3. The learning setup seems problematic:
3a. The model may overfit for the genomic sequences that contain TF binding sites as it has never seen genomic sequences without TF binding sites (the genomic sequences that don’t have CHIP peaks are discarded from the dataset). Performance for genome wide scans should definitely include those to assess accuracy.
3b. The train/validation/test are defined by chromosome. There does not seem to be any screening for sequence similarity (e.g. repetitive sequences, paralogs). This may inflate performance, especially for more complicated models which may be able to “memorize” sequences better.
4. The paper claims to have 4 major contributions. The details of second claim that the prototype matching loss learns motif like features is not explained anywhere in the paper. If we look at the actual loss function equation (12), it penalizes the difference between the label and the similarity score but the prototypes are not updated. The fourth claim about the biological relevance of the network is not sufficiently explored. The authors show that it learns co-bindings already known in the literature which is a good sanity check but does not offer any new biological insight.  The actual motifs or the structure of their relations is not shown or explored.
5. PWN offers only marginal improvement over the CNN networks

---

> ### Author Response · Authors · 2018-01-05
> **Response to 5.**
>
> The PMN model does only offer a marginal improvement over CNNs. However, we believe that our architecture models the biology better, which could lead to new insights. This is similar to the marginal improvements of DanQ over DeepSEA, but the DanQ model was better fitting for the biology.

---

> ### Author Response · Authors · 2018-01-05
> **Response to 4.**
>
> We realize that we did not explain the idea of prototypes very well in the original manuscript. While we said the the prototypes learn motif-like features, they actually learn high level abstract representations summarizing patterns of each TF  through an embedding vector. Since we use a lookup table for the prototypes, they are in fact updated via gradient descent on each update.
>
> It is true that we did not add any new biological insight, but our goal for this paper was rather to design a prediction model. In future work, we plan to find new insights which can be used by biologists.

---

> ### Author Response · Authors · 2018-01-05
> **Response to 3.**
>
> 3a. We thank the reviewer for pointing out a valid concern in our dataset. We felt that for this experiment, running on only windows with at least one ChIP-seq peak was sufficient, especially due to runtime constraints of including more windows, but we will include completely negative windows in future experiments.
>
> 3b. We would like to thank the reviewer for noting another important concern with out dataset in that there is no screening for sequence similarity. However, since this method of dividing the splits up by chromosomes was done in previous datasets (DeepSEA, ENCODE DREAM), we adopted the same methodology.

---

> ### Author Response · Authors · 2018-01-05
> **Response to 2.**
>
> The DanQ method (Quand and Xie) applies a bidirectional LSTM on top of the CNN outputs, which finds dependencies among motifs at the sequence level (i.e., among different sequence positions). Our method is concerned with finding dependencies among TFs at the output level  (i.e., among different labels), but we will compare to that method to show that modelling TF interactions is stronger than modelling motif interactions. DanQ does not report actual AUC values, but rather the improvements over DeepSEA, so we need to implement our own CNN+LSTM model similar to DanQ for future work.
>
> We would like to thank the reviewer for noting related works which we did not cite. Our primary goal was to compare against state of the art deep learning methods which do not incorporate co-binding, but we realize that we should be citing related non deep learning works.

---

> ### Author Response · Authors · 2018-01-05
> **Response to 1.**
>
> We thank the reviewer for pointing out that there should be a better comparison against the prior deep learning works in TFBS prediction. Our baseline CNN was the same architecture (with slightly different hyperparameters) as DeepBind, DeepSEA and Basset. DeepBind and Basset are for single task (one label per model), but we realize that we should run our model on the DeepSEA dataset which is multi-label for TFs, Histone Modifications and DNA accessibility. However, our model was designed to model the dependencies among TF binding, which may be skewed in the DeepSEA dataset which also has HM and DNase outputs.

---

### Official Review · AnonReviewer1 · 2017-11-28
**This work proposes an approach for transcription factor binding site prediction using a prototype matching network. While the approach is interesting, the work needs further improvements to be appealing to the ICLR audience.**

**Rating:** 5
**Confidence:** 4

**Review:**

This work proposes an approach for transcription factor binding site prediction using a multi-label classification formulation. It is a very interesting problem and application and the approach is interesting.

Novelty:
The method is quite similar to matching networks (Vinyals, 2016) with a few changes in the matching approach. As such, in order to establish its broader applicability there should be additional evaluation on other benchmark datasets. The MNIST performance comparison is inadequate and there are other papers that do better on it.
They should clearly list what the contributions are w.r.t to the work by Vinyals et al 2016.
They should also cite works that learn embeddings in a multi-label setting such as StarSpace.

Impact:
In its current form the paper seems to be most relevant to the computational biology / TFBS community. However, there is no comparison to the exact networks used in the prior works DeepBind/DeepSea/DanQ/Basset/DeepLift or bidirectional LSTMs. Further there is no comparison to existing one-shot learning techniques either. This greatly limits the impact of the work.

For biological impact, a comparison to any of the motif learning approaches that are popular in the biology/comp-bio community will help (for instance, HOMER, FIMO).

Cons:
The authors claim they can learn TF-TF interactions and it is one of the main biological contributions, but there is no evidence of why (beyond very preliminary evaluation using the Trrust database). Their examples are 200-bp long which does not mean that all TFs binding in that window are involved in cooperative binding. The prototype loss is too simplistic to capture co-binding tendencies and the combinationLSTM is not well motivated. One interesting source of information they could tap into for TF-TF interactions is CAP-SELEX (Jolma et al, Nature 2015).

One of the main drawbacks is the lack of interpretability of their model where approaches like DanQ/DeepLift etc benefit. The PWM-like filters in some of the prior works help understand what type of sequence properties contribute to binding events. Can their model lead to an understanding of this sort?

Evaluation:
The empirical evaluation itself is not very strong as there are only modest improvements over simple baselines. Further there are no error-bars etc to indicate the variance in their performance numbers.
It will be useful to have a TF-level performance split-up to get an idea of which TFs benefit most.

Clarity:
The paper can benefit from more clarity in the technical aspects. It is hard to follow for anyone not already familiar with matching networks. The objective function, parameters need to be clearly introduced in one place. For instance, what is y_i in their multi-label framework?
Various choices are not well motivated; for instance cosine similarity, the value of hyperparameter epsilon.
The prototype vectors are not motif-like at all -- can the authors motivate this aspect better?

Update: I have updated my rating based on the author rebuttal

---

> ### Author Response · Authors · 2018-01-05
> **Clarity**
>
> We thank the reviewer for noting unclear technical aspects. y_i is the ground truth label (0 or 1) for TF i. y should be denoted bold for a vector, which we have changed.
>
> We chose cosine similarity because we wanted a distance measure which mapped the similarity between 0 and 1 (since it’s multi-label). We tried squared Euclidean with a margin loss, but it did not perform as well. We realize that we did not explain this well in the original draft.
>
> We chose epsilon=20 because we wanted a large enough epsilon so that the softmax output would be between 0 and 1. Since the max value of cosine similarity is 1, this would result in sigmoid(1) = ~0.7 . Thus we chose epsilon=20 so that sigmoid(1*20) = ~1.
>
> The prototype vectors are not in fact like traditional motifs, but rather high level hidden representations of the TFs themselves. The CNN filters extract the individual motifs, but the prototypes are higher-level summary representations of each TF.

---

> ### Author Response · Authors · 2018-01-05
> **Evaluation**
>
> We agree that more robust evaluations should be added to convince readers of our method. We did include the pairwise t-test among TFs which showed that our method significantly outperformed baselines. However, showing TF level performance and variance in metrics will greatly help.

---

> ### Author Response · Authors · 2018-01-05
> **Cons**
>
> It is true that not all TFs in a window are involved in cooperative binding. However, we believe that our model handles this by not updating the matching score after iterating over the other TFs using the combinationLSTM. Although we have not experimentally verified this, we believe that the prototype loss does capture co-binding tendencies because of the fact that using the loss function after all the hops performs better than the loss function without iterative hops.
>
> We would like to thank the reviewer for pointing out the CAP-SELEX paper, which is an extremely good resource for validating our method. We plan to do this for future work.
>
> We would like to thank the reviewer for noting the lack of interpretability of our model, which is something that we realize is an important factor in computational biology. As previously noted, we believe that our CNN extracts similar motif features as in DeepBind, DanQ, and Basset. However, we will validate this in future work.

---

> ### Author Response · Authors · 2018-01-05
> **Impact**
>
> We thank the reviewer for pointing out that there should be a better comparison against the prior deep learning works in TFBS prediction. Our baseline CNN was the same architecture (with slightly different hyperparameters) as DeepBind DeepSea and Basset, but we realize that we should compare against those exact methods. The DanQ method applies a bidirectional LSTM on top of the CNN outputs, which finds dependencies among motifs (i.e., among different sequence positions). Our method is concerned with finding dependencies among TFs (i.e., among different labels), but we should compare to that method. If the task is for single label few shot learning (as in Vinyals et al. and Snell et al.) then our method is very similar to the previous, with the one major difference of learning the support set as opposed to using the images directly or a mean of the images.
>
> We decided not to compare against traditional motif learning methods such as HOMER and FIMO because it has been shown that the CNN filters can find related motifs (e.g. Alipanahi et al. 2015, Kelley et al. 2015). Since the first step of our method is a CNN, we assume that these filters learn these first-order motifs. We plan to validate this using the same method as Alipanahi et al. in future work. In addition, Alipanahi et al. showed that a basic 1-layer CNN model can outperform the baseline MEME-ChIP approach (Machanick & Bailey 2011), which uses the traditional position weight matrix motif approach. Thus, we did not also compare against MEME-ChIP for accuracy.

---

> ### Author Response · Authors · 2018-01-05
> **Novelty**
>
> We have added a table to compare our method to the similar work in Vinyals et al. 2015 and Snell et al. 2017. The main contribution of our work over the previous two is that we extend those methods to a large-scale and multi-label task. We are currently working on adding more benchmark datasets in the large-scale and multi-label tasks so that we can further prove our method. We have also added a short review of the recent StarSpace paper in the previous multi-label works section.

---

### Decision · Program_Chairs · 2018-01-29
**ICLR 2018 Conference Acceptance Decision**

**Decision:**

Reject

**Comment:**

This paper proposes an approach for predicting transcription factor (TF) binding sites and TF-TF interaction.  The approach is interesting and may ultimately be valuable for the intended application.   But in its current state, the paper has insufficient technical novelty (e.g. relative to matching networks of Vinyals 2016), insufficient comparisons with prior work, and unclear benefit of the approach.  The reviewers also had some concerns about clarity.